# Mechanical Properties and Failure Behavior of Dry and Water-Saturated Foliated Phyllite under Uniaxial Compression

**DOI:** 10.3390/ma15248962

**Published:** 2022-12-15

**Authors:** Guanping Wen, Jianhua Hu, Yabin Wu, Zong-Xian Zhang, Xiao Xu, Rui Xiang

**Affiliations:** 1State Key Laboratory of Safety and Health for Metal Mines, Sinosteel Maanshan General Institute of Mining Research Co., Ltd., Maanshan 243000, China; 2School of Resources and Safety Engineering, Central South University, Changsha 410083, China; 3Oulu Mining School, University of Oulu, 90570 Oulu, Finland; 4Zijin School of Geology and Mining, Fuzhou University, Fuzhou 350108, China; 5Changsha Institute of Mining Research Co., Ltd., Changsha 410012, China

**Keywords:** anisotropic rock, phyllite, crack evolution, failure modes, DIC method, AE parameter analysis

## Abstract

Phyllite is widely distributed in nature, and it deserves to be studied considering rock engineering applications. In this study, uniaxial compression tests were conducted on foliated phyllite with different foliation angles under dry and water-saturated conditions. The impacts of water content and foliation angle on the stress–strain curves and basic mechanical properties of the Phyllite were analyzed. The experimental results indicate that the peak stress and peak strain decrease first and then increase with increasing foliation angle as a U-shape or V-shape, and the phyllite specimens are weakened significantly by the presence of water. Moreover, an approach with acoustic emission, digital image correlation, and scanning electron microscopic is employed to observe and analyze the macroscopic and mesoscopic failure process. The results show that tensile microcracks dominate during the progressive failure of phyllite, and their initiation, propagation, and coalescence are the main reasons for the failure of the phyllite specimens. The water acts on biotite and clay minerals that are main components of phyllite, and it contributes to the initiation, propagation, and coalescence of numerous microcracks. Finally, four failure modes are classified as followed: (a) for the specimens with small foliation angles α = 0° or 30° (Saturated), both shear sliding and tensile-split across the foliation planes; (b) for the specimens with low to medium foliation angles α = 30° (Dry) or 45°(Saturated), shear sliding dominates the foliation planes; (c) for the specimens with medium to high foliation angles α = 45° (Dry) or 60°, shear sliding dominates the foliation planes; (d) for the specimens with high foliation angles α = 90°, tensile-split dominates the foliation planes.

## 1. Introduction

As a typical metamorphic rock, phyllite is encountered frequently in mining and geotechnical engineering, e.g., in tunneling, underground construction, mining, oil and gas extraction, etc. Practical operations in tunneling and underground mining show that a large number of foliation planes often appear in phyllite, and these foliation planes have a considerable influence on the mechanical properties and failure behavior of the phyllite [1,2,3,4,5]. In addition, various geological disasters such as roof caving, rib spalling, and rockburst are relevant to foliation planes that bring great difficulty for rock excavation and support [6,7,8,9,10].

To investigate the failure behavior of foliated or layered rock materials, many scholars derived a number of constitutive laws and failure criteria for transversely isotropic rocks [11,12,13,14]. These constitutive laws and failure criteria agreed well with the experimental data of many investigators in terms of prediction of strength and deformation, but they were difficult to be used to characterize crack evolution during the loading process. Accordingly, substantial studies were carried out on the strength, deformation characteristics, and failure behavior of transversely isotropic rocks by both experimental and numerical methods [15,16,17,18]. Tien et al. [19] investigated 3D macroscopic-fractured surfaces of simulated transversely isotropic rocks by reconstructing unrolled images from a rotary scanner and divided the failure modes into four kinds: sliding failure along discontinuities, tensile fracture across the discontinuities, tensile-split along the discontinuities, and sliding failure across the discontinuities. Yin and Yang [20] studied three kinds of transversely isotropic rock-like specimens with different bedding thickness ratios under conventional triaxial compression and divided the failure modes into three kinds: tensile-split along the core axes (0°, 15°, and 30°), shear-sliding along the bedding plane (45°, 60°, and 75°), and split along the vertical bedding plane (90°). Furthermore, the development of the Digital Image Correlation method (DIC) [21,22,23], Acoustic Emission techniques (AE) [24,25,26,27], and Scanning Electron Microscopic observation (SEM) was found to be suitable for capturing the progressive failure process and understanding the failure mechanism of the rock. The aforementioned studies show that the failure mode transforms from tensile splitting across discontinuities to shear slip along discontinuities, and finally to tensile splitting along discontinuities with increasing foliation angle.

In addition to foliation angle, water content has a significant impact on the mechanical properties and failure behavior of phyllite. Previous studies indicate that the presence of water significantly weakens the mechanical properties of the rock such as its strength, rigidity, and brittleness [28]. In the saturation process, microcracks are filled with water, and crack tips are the most active areas of the water-rock reaction, which may decrease the critical stress intensity factor [29,30]. Furthermore, phyllite is a compact lustrous metamorphic rock. The rock is always rich in mica, chlorite, and quartz and it possesses the lepido granoblastic texture. Some changes may happen when phyllite is saturated such as clay mineral expansion, pore filling, non-uniform deformation, and calcite dissolution, respectively [31,32]. All of these changes result in the weakness of physical-mechanical properties and the generation of new defects like microcracks. 

Though various isotropic rocks were studied, many new techniques in laboratory testing have made it possible to systematically investigate the coupling effects of foliation angle and water content on the mechanical properties and failure behavior by using an approach of AE, DIC and SEM techniques for foliated phyllite under uniaxial compression.

## 2. Materials and Methods

### 2.1. Rock Specimens

The rock samples used in this study are foliated phyllite, taken from Longling county, Yunnan province, China. The X-ray diffraction result shows the mineral composition of the phyllite consisting of approximately 53.8% biotite, 22.4% kaolinite, and 16.4% quartz, as shown in Figure 1. The SEM observation of the phyllite specimens indicates that foliations are parallel to one another with a thickness of from 20 to 140 μm, and a large number of microcracks and micropores are scattered in foliation planes.

According to ISRM suggested methods [33,34], a large phyllite block from the field was manufactured into d50 mm × 100 mm cylindrical specimens having five foliation angles 0°, 30°, 45°, 60°, and 90°. These specimens were divided into two parts: dry groups and water-saturated groups. The dry groups were placed in a drying oven for 24 h until the weight of each specimen no longer decreased, and the water-saturated groups were placed in a vacuum water-filling device for 48 h until the weight of each specimen no longer increased. Moreover, all specimens were sealed with plastic wrap once the process of drying or saturation was completed. To select intact specimens, serials of density and wave velocity tests were carried out using the prepared phyllite specimens to remove the specimens with macro-fractures induced by specimen preparation. In this study, all properties had been tested twice for obtaining reliable results. The physical properties of the intact phyllite specimens are tabulated in Table 1. 

In Table 1, the average density of phyllite in the dry and water-saturated conditions is 2.762 g/cm^3^ and 2.775 g/cm^3^, respectively. The density of phyllite in dry conditions was significantly lower than that in water-saturated conditions, indicating that phyllite had a good water-absorbing ability. Furthermore, the water-absorbing ability is significantly affected by foliation angles, and the water-absorbing ability with different foliation angles from strong to weak is 0°, 60°, 30°, 45°, and 90°. Furthermore, the wave velocity is also significantly influenced by foliation angle and water content. When the foliation angle increased from 0° to 90°, the average wave velocity of specimens under dry conditions increased from 3750 to 5942 m/s (58.5% increment). The average wave velocity of specimens under dry conditions is lower than that of under water-saturated conditions. From the physical features, the phyllite specimens in this study are of the typical isotropic rocks and have a good water-absorbing ability.

### 2.2. Testing Setup and Methods

The testing setup consists of a loading system, a DIC system, and an AE system, as shown in Figure 2. Uniaxial compression tests were conducted on a new SANS electro-hydraulic servo-controlled rigidity testing machine and a loading control system DCS-200. According to ISRM Suggested methods for rock characterization, testing, and monitoring [33], the stress control mode was adopted in the uniaxial compression tests and the stress loading rate was set at 0.5 MPa/s. In order to record the crack evolution of the foliated phyllite under uniaxial compression, both the Digital Image Correlation (DIC) method and acoustic emission (AE) technology were conducted. The DIC system consisted of an industrial camera with a resolution of 2448 × 2048 pixels and a frame rate of 15 frames per second, two LED lamps for supplemental lights, and a computer with GOM correlate software for image collection and processing. The AE system was composed of a MICRO-II-32 all-digital AE workstation produced by American Physical Acoustics Corporation, four amplifiers with a 40 dB threshold, and four sensors fixed on both sides of the specimens.

## 3. Mechanical Behavior Analysis

### 3.1. Stress–Strain Curves

Figure 3 shows the stress–strain curves of foliated phyllite with different foliation angles under uniaxial compression. Compared with the characteristics of typical stress-strain curves of low-porosity rocks [35,36], the stress–strain curves of the phyllite are different in three aspects: (1) fluctuating at the elastic deformation stage; (2) a lack of a clear stable and unstable crack growth stage; (3) dropping rapidly after the peak stress but maintaining a certain residual strength. The stress–strain curves of the phyllite specimens with various foliation angles were apparently different from those with various water contents. As shown in Figure 3, at the pore compaction stage, the compacting ability of the specimens decreased with increasing foliation angle but increased with increasing water content. At the elastic deformation stage, the axial strain at α = 0° and α = 90° was apparently larger than that at α = 30°, 45°, and 60°, and this case was more obvious in the water-saturated specimens. At the stable crack growth stage and unstable crack growth stage, the axial stress increased slowly with increasing axial strain for a short loading duration, especially for specimens under dry conditions. At the post-peak stage, the axial strain at α = 0° dropped rapidly after peak load but maintained a certain residual strength.

### 3.2. Mechanical Properties

The basic mechanical properties of phyllite specimens such as uniaxial compressive strength, peak axial strain and elastic modulus were obtained. Figure 4 indicates the mechanical properties of the dry and saturated phyllite specimens. To define the anisotropy degree of foliated phyllite specimens, the ratios between the largest and smallest values of different mechanical properties are employed such as *σ_c_^max^/σ_c_^min^*, ε_*c*_^*max*^/ε_*c*_^*min*^, and *E^max^/E^min^*. 

As illustrated in Figure 4a, the average strength *σ_c_* of foliated phyllite varies with foliation angle α in a U-shape, and the maximum strength is at α = 90° while the minimum strength is at α = 30°. The foliation angle has a significant effect on the strength: For dry specimens, the maximum strength is 116.6 MPa at α = 90° and the minimum strength is 33.7 MPa at *α* = 30°, and *σ_c_^max^/σ_c_^min^* is approximately equal to 0.29; For saturated specimens, the maximum strength is 104.6 MPa at α = 90° and the minimum strength is 9.6 MPa at *α* = 30°, and *σ_c_^max^/σ_c_^min^* is approximately equal to 0.09. Furthermore, the presence of water significantly reduces the strength of the phyllite, and the reduction is related to the foliation angle. For specimens with different foliation angles, the maximum strength reduction is 71.4% at *α* = 30°, the least strength reduction is 10.3% at *α* = 90°, and the average strength reduction is 36.0%.

Figure 4b shows that the average strain *ε_c_* of foliated phyllite varies with the foliation angle α in a V-shape, and the maximum strain is 4.9 × 10^−3^ at α = 0° or 90° while the minimum strain is at α = 30°. The foliation angle has a significant effect on peak axial strain: for dry specimens, the maximum strain is 4.9 × 10^−3^ at α = 0° and the minimum strain is 2.3 × 10^−3^ at *α* = 30°, and *ε_c_^max^*/*ε_c_^min^* is approximately equal to 0.47; for saturated specimens, the maximum strain is 4.4 × 10^−3^ at α = 90° and the minimum strain is 1.8 × 10^−3^ at *α* = 30°, and *ε_c_^max^*/*ε_c_^min^* is approximately equal to 0.41. Furthermore, the presence of water significantly reduces the strain of phyllite, and the reduction is related to the foliation angle. For specimens with different foliation angles, the maximum strength reduction is 59.5% at α = 60°, the least strength reduction is 12.6% at α = 90°, and the average strength reduction is 33.6%.

Figure 4c displays that the average elastic modulus *E* of foliated phyllite varies with foliation angle α in a wavy shape. It is obvious that the presence of water has a significant influence on elastic modulus of phyllite: for specimens of dry groups, the elastic modulus first increases, then decreases and finally increases again with increasing foliation angle; for specimens of saturated groups, the elastic modulus first decreases, then increases, and then decreases and finally increases again with increasing foliation angle. Generally, the elastic modulus of dry phyllite is large than that of saturated phyllite, for example, the elastic modulus reduction is 76.9% at α = 30°. However, there are some special situations, e.g., the elastic modulus increment is 69.9% and 43.9% at α = 45° and α = 90°, respectively. Overall, the discreteness of elastic modulus of dry phyllite is less than that of saturated phyllite for different foliation angles.

## 4. Crack Evolution Behavior Analysis

### 4.1. Macro-Crack Evolution Behavior

According to the axial stress–strain curves of foliated phyllite, a typical entire crack evolution process is shown in Figure 5. To obtain a set of representative pictures of maximum principal strain fields during the whole loading stage, some specific photos at the moments corresponding to special loading stages were chosen. These photos were imported to the GOM-correlated software and the surface strain fields were analyzed. In addition, the characteristics of crack evolution were obtained by analyzing the counts rate and accumulated energy of AE generated by microcracks in rocks during uniaxial compression.

Figure 6 demonstrates the crack evolution process of dry phyllite with 0° foliation angle at different loading times. A shear sliding crack initiates from the end of specimen and extends obliquely downward to the middle area of the specimen. Then, a tensile splitting crack, connected to the initial shear sliding crack, vertically propagates downwards to the inside specimen. Meantime, another shear sliding crack initiates from the connection position of shear sliding crack and tensile splitting crack and extends obliquely upward to the end of specimen. Afterwards, an approximately inverted conical failure surface appears and further promotes the initial tensile splitting crack vertically downward and finally causes the instable failure of the specimen. Significantly, the initiation and extension of cracks are accompanied by the evident variations of AE events rate and accumulated AE energy. Though AE count peaks appear several times and the cumulative counts rise step by step, the largest peak of AE events rate and the maximum accumulated AE energy occurs at the failure moment of the specimen. The crack evolution process of water-saturated phyllite at α = 0° and 30° was similar to that of dry phyllite α = 0°, but there are also significant differences between them. For example, the extension of the initial shear sliding crack is steeper and the propagation of the tensile splitting crack easily bends and turns to a foliation plane. In addition, the value of AE events rate and accumulated AE energy of water-saturated phyllite are over one order of magnitude smaller than those of water-saturated phyllite. These results show that water has a significant influence on both a reduction in the intensity of the AE signal generated by rock failure and a weakening of the signal reception by hindering signal propagation.

The crack evolution processes of dry phyllite at α = 30° and water-saturated phyllite at α = 45° were similar. Figure 7 illustrates the crack evolution process of water-saturated phyllite with 45° foliation angle at different loading times. Several shear sliding cracks initiate from the end of specimen and extend obliquely downward to the position around the middle area of specimen. Then another shear sliding crack, connected to the initial shear sliding crack, extends obliquely downward to one side of specimen. Afterwards, a triangular sliding block appears in one side of the specimen and finally causes the instable failure. Moreover, a sharp increase in AE events rate and an accumulated AE energy would occur at the moment when the stress suddenly drops and the larger the stress drops, the greater the AE events rate and accumulated AE energy increment are. The crack evolution process of dry phyllite at α = 30° was similar to that of water-saturated phyllite α = 45°.

Figure 8 shows the crack evolution process of water-saturated phyllite with 60° foliation angle at different loading times. A shear sliding crack initiates from the end of the specimen and extends obliquely downward. Then, several shear sliding cracks propagate from the end of the specimen, which is near to and parallel to initial shear sliding crack. Moreover, the propagation path of the shear sliding crack is located in one of foliation planes and causes the ultimate instable failure of specimen. The crack evolution processes of water-saturated phyllite α = 60° and dry phyllite at α = 45° and 60° are similar, but two differences still exist in them: (1) the crack propagation along foliation planes is affected by the foliation angle; (2) the water content contributes to decrease the resistance of crack propagation so that few shear sliding crack initiates from the end of the specimen and extends obliquely downward along foliation planes in the water-saturated specimen. 

Figure 9 shows the crack evolution process of dry phyllite with 90° foliation angle at different loading times. Several tensile splitting cracks initiate from the end of the specimen and extend vertically downward to the other end of the specimen. Especially, the outside tensile splitting crack extends faster than the inside one, causing the block spalling of specimen step by step. The crack evolution processes of water-saturated phyllite α = 90° and dry phyllite at α = 90° were similar except that the number of tensile splitting cracks is few. 

### 4.2. Micro-Crack Failure Modes

The DIC method can effectively identify the macro-crack evolution on the surface of specimens. Additionally, Acoustic Emission (AE) can be used to study the real-time formation and growth of local failure in rock materials by means of AE event counts (Accumulated AE event counts), AE energy (Accumulated AE energy), crack classification, AE peak frequency, and AE position. AE event counts and AE energy can be adopted in analyzing macro-crack evolution. In addition, AE parameter analysis can be used to classify tensile crack mode and shear/mixed crack mode at a microscopic scale in rock mechanics. As illustrated in Figure 10, the waveform of the tensile events is mainly in the form of longitudinal waves, and the RA value is low in one hit event. On the contrary, the RA value of shear wave resulting from shear events is usually higher. Thus, many scholars put forward adopting the ratio of RA value to the Average Frequency (AF) as a classification criterion and the slope of the transition line between tensile crack and shear crack is commonly decided by the rock type. Based on previous studies [37,38,39,40], the optimal ratio of AF and RA is approximately from 100 to 500 for brittle rock materials subjected to compression loading, and it was adopted as 100 in this study in the light of the phyllite lithology.

Figure 11 shows typical scatter diagrams between AF and RA values for foliated phyllite with different foliation angles under dry and water-saturated conditions. The AF-RA points are divided into two groups by the transition line. The upper groups (red) are located in the domain where AF value is higher and RA value is lower, which is consistent with the feature of tensile crack. Similarly, the lower groups (black) are located in the domain where AF value is lower and RA value is higher, which is consistent with the feature of shear/mixed crack. Furthermore, the intensity degree of AF-RA points reflects the crack scale during rock deformation. Among the specimens with different foliation angles, the specimen with 30° foliation angle yields the minimum cracking scale and the one of 0° foliation angle does the maximum cracking scale. Compared with the dry specimens, the saturated specimens yield relatively lower cracking scale.

Figure 12 shows the proportions of shear and tensile cracks for foliated phyllite with different foliation angles. For the dry specimens, the proportions of shear and tensile cracks in phyllite are 19.87% and 80.13% on average, respectively. Additionally, the maximum proportion of shear cracks is 31.88% at α = 0°, and the minimum proportion of shear cracks is 13.17% at α = 45°. For the saturated specimens, the proportions of shear and tensile cracks in phyllite are 13.43% and 86.57% on average, respectively. The maximum proportion of shear cracks is 18.36% at α = 0°, and the minimum proportion of shear cracks is 8.84% at α = 60°. Overall, the proportion of tensile cracks is almost 2 to 7 times of that of shear cracks, which is much different from the crack classification results of other rock types in previous studies [37]. For example, the proportions of shear cracks in marble and fine-grained granite are 66.81% and 84.79% on average, respectively. These results show that the tensile microcracks are more easily developed than shear microcracks, and the initiation, propagation, and coalescence of tensile microcracks are the main reasons for the failure of the phyllite specimens.

### 4.3. Ultimate Failure Mode

Previous studies had put forward that the failure mechanism of layered rock such as sandstone and shale is mainly controlled by the weak cementation between the foliation or bedding planes [18,30]. As illustrated in Table 2, the failure mode of foliated phyllite mainly depends on the foliation angle and is significantly influenced by water. When the foliation angle was small, the shear sliding cracks were developed through the foliation plane first, and they would not further extend once the frictional resistance was large enough. Then, two cases would appear: at α = 0°~30°, a shear sliding crack propagates from the inner tip of initial shear sliding cracks to the top of the specimen; at α = 30°~45°, a shear sliding crack propagates from the inner tip of initial shear sliding cracks to the side of the specimen. When the foliation angle was larger (45°–60°), the shear sliding cracks grew along the foliation plane so that they could not be stopped by the frictional resistance. When the foliation angle was a nearly vertical plane (90°), the tensile splitting cracks extended along the foliation plane firstly and destroyed the specimen gradually. Overall, the failure of foliated phyllite tended to change from shear sliding mode to tensile splitting mode with increasing foliation angle, and it was more and more closely related to the foliation plane. 

Figure 13 shows four typical failure modes of foliated phyllite and the failure sketches of two adjacent tunnels with different foliation angles. Where T1 shows tensile splitting cracks developed along the foliation plane, T2 does tensile splitting cracks developed through the foliation plane, S1 indicates shear slide cracks developed along the foliation plane, and S2 does shear slide cracks developed through the foliation plane. The first is the mixed mode (sliding + splitting). A tensile splitting crack, initiated from the intersection of two shear sliding cracks, develops downward across the foliation plane. The second is the sliding mode. The failure of the specimens in this mode presented a triangular sliding block on one side of the specimen caused by the intersection between one shear sliding crack developed along the foliation plane and another shear sliding crack developed through the foliation plane. The third is also the sliding mode. The failure of the specimens in this mode was caused by one shear sliding crack that developed along the foliation plane and got through from one side of the specimen to the other. The last one is the splitting mode. This mode presented multiple tensile splitting cracks developed along across the foliation plane and divided the specimens into several long columns. Finally, these long columns gradually were buckled.

### 4.4. Microscopic Characteristics Analysis in Typical Fracture Surface

Previous studies indicate that microscopic morphological characteristics are related to crack initiation, crack propagation, and fracture modes [41]. To reveal the connection between them, the Scanning Electron Microscopic (SEM) technique was adopted to observe the morphological characteristics of typical failure fracture. 

According to Mohr–Coulomb criterion, the shear failure of intact specimens often occurs along the direction of the maximum shear stress subjected to uniaxial compression loading. However, previous studies indicate that the foliation plane and water content have a great influence on the failure of rock specimens. As shown in Figure 14, the orientation of maximum principal stress (σ_1_) is always acting vertically downward, and the orientation of maximum shear stress (τ_max_) is only dependent on the internal friction angle of the rock material. With increasing foliation angle, the angle between the foliation plane and maximum shear stress (τ_max_) gradually decreases at first to zero and then increases, and the angle between the foliation plane and maximum principal stress (σ_1_) decreases from 90° to 0°. 

Figure 15 shows that the SEM scanning characteristics of four typical shear fracture surfaces of phyllite. When the foliation angle is close to 0°, the component of maximum principal stress (σ_1_) vertical to the foliation plane is larger than that of maximum principal stress (σ_1_) horizontal to the foliation plane. In this condition, the foliation plane hardly affects the initial failure process. As illustrated in Figure 15a, the whole shear fracture surface is relatively complete but partially broken with the flaky mineral particles that mostly present alternating foliations contact, and the surface appears with irregular jagged characteristics because of the shear failure in intergranular and trans-granular cracks. Overall, the whole surface is rough, but the area between two adjacent foliations is much smoother than that in other areas. Furthermore, the irregular jagged fracture surfaces are also flat but the orientation of them is different. Nevertheless, there is also a slight difference between foliated phyllite and intact rock materials, and the difference lies in the crack propagation path transforming from shearing slide along maximum shear stress to tensile-split along maximum principal stress. 

With an increasing foliation angle, the component of maximum principal stress (σ_1_) horizontal to the foliation plane is large enough to shearing slide along the foliation plane. As illustrated in Figure 15b–d, the whole shear fracture surface almost is located in the cleavage plane and is relatively flat. Significantly, the fracture surface damage is closely related to the component of maximum principal stress (σ_1_) vertical to the foliation plane, which decreases with the increasing foliation angle. For specimens with a 45° foliation angle, the shear sliding between different thin foliation planes resulted in a great number of river-like traces, which were approximately perpendicular to the shear direction and are randomly scattered on the step-shaped fracture surface. Furthermore, the shear failure between adjacent foliation planes brought about numerous debris appearing in the cleavage edge. For specimens with a 60° foliation angle, some river-like traces were also approximately perpendicular to the shear direction, but the number of traces was smaller than that of specimens with a 45° foliation angle. Numerous pieces of debris appear in both the edge and middle of the cleavage, reflecting that the failure of the foliation plane is influenced by both shear sliding and splitting tensile.

Figure 16 shows that the characteristics of two typical tensile splitting fracture surfaces of phyllite from SEM observation. When the foliation angle is close to 90°, the maximum principal stress (σ_1_) is approximately parallel to the foliation plane, which makes the tension at the foliation plane large enough to develop splitting tensile cracks along the foliation plane and lead to failure of the specimen. As shown in Figure 16, the fracture surface across the foliation plane is relatively rough but that along the foliation plane is flat. There are many broken flaky mineral particles in the fracture surface across the foliation plane, having the characteristics of intergranular or trans-granular tensile failure. Meanwhile, these areas between two adjacent foliations are much smoother than other areas. For the fracture surface along the foliation plane, the tensile failure between adjacent foliation planes brought about the appearance of numerous debris in the cleavage. The results show that the foliation plane in phyllite plays a vital role in changing failure behavior and affecting the microscopic morphology of the fracture surface.

Energy Dispersive Spectroscopy (EDS) has been widely used in the mineral composition analysis of rock materials with the advantage of easy operation, quick analysis, and low cost. By scanning the region within the 1 μm fracture surface, the element from Beryllium (Be) to Uranium (U) could be identified [42,43]. In this study, four rock fractures were selected from the parallel and vertical surface of the foliation plane in specimens under dry and water-saturated conditions. All experiments were repeated twice by changing the scanning region and magnification. As illustrated in Table 3, the contents of Sodium, Magnesium, and Calcium are more extensive in specimens under dry conditions, and the content of potassium is more extensive in specimens under water-saturated conditions. This result indicates that the amount of biotite contained in phyllite participated in the hydration reaction, resulting in the Potassium ions (Na^+^) in biotite being replaced by other macromolecules or the ions of Sodium (Na^+^), Magnesium (Mg^+^), and Calcium (Ca^+^). Due to the radii of these replaced macromolecules or ionic being larger than the radius of Potassium ions (Na^+^), some tensile cracks initiate and propagate from biotite cleavage, and then small fragments gradually divorce from the edge of biotite cleavage. Thus, the presence of water could contribute to accelerating the evolution of micro-cracks. Furthermore, the contents of Silicon, Calcium, and Iron are more extensive on the fracture surface that is vertical to the foliation plane, and the contents of Aluminium and Titanium are more extensive on the fracture surface parallel to the foliation plane. This indicates that the biotite on the foliation plane is larger than that on the vertical foliation plane and the quartz on the foliation plane is less than that on the vertical foliation plane, reflecting that the mineral composition distribution of phyllite is anisotropic.

For water-saturated phyllite specimens, the internal micropores and microcracks are full of water so that all minerals are in full contact with the water. Owing to the clay minerals having a large specific surface area and good hydrophilicity, the thickness of the water film gradually increases, accompanied by the expanding mineral particles and the weakness of adhesion between particles, which results in the initiation and propagation of numerous microcracks (Figure 17). Furthermore, the distribution of mineral composition and micro defects in phyllite is inhomogeneous, which may cause extra internal stress and inhomogeneous deformation between minerals particles. 

## 5. Discussion

Extensive studies have been performed to study anisotropic rocks [1,19,20,44,45,46,47]. Figure 18 indicates that the uniaxial compressive strength of anisotropic rock changes with foliation angle or bedding angle α in a U-shape, shoulder shape, or wave shape. In this study, the strength of foliated phyllite with multiple parallel macro weak planes varies with foliation angle α in a U-shape, and the maximum strength is at α = 90° while the minimum strength is at α = 30°. In particular, the strength of the specimen at α = 45° large than that at α = 30° or α = 60°, and in this sense, the curve of strength and foliation angle is in a wave shape.

Furthermore, many scholars [1,4,19,20] have put forward many classifications of ultimate failure mode for anisotropic rock, and typical failure modes have been listed in Table 4. For specimens of S-D-45, S-D-60, and S-W-60, the shear cracks along discontinuities (S1) cause the overall failure, which is similar to category SD in Tien et al. For specimens of S-D-90 and S-W-90, the tensile-split along discontinuities (T1) causes the overall failure, which is similar to category TD in Tien et al. However, there exist two types different from previous studies in terms of the failure process. Both tensile fracture across discontinuities and sliding failure across discontinuities were not found in the study. For specimens of S-D-0, S-W-0, and S-W-30, a combination of sliding failure across discontinuities and tensile fracture across discontinuities (M1) causes the overall failure. For specimens of S-D-30 and S-W-45, a combination of sliding failure along discontinuities and sliding failure across discontinuities (M2) causes the overall failure.

Future research should carefully consider the effects of water content and foliation angle on foliated phyllite in practical applications. Moreover, theoretical analysis and numerical simulation are necessary to investigate the damage processes and crack evolution.

## 6. Conclusions

The following conclusions can be drawn from the experiments in this study:

(1) The axial stress–strain curves of foliated phyllite have at least one or more features as follows: (a) fluctuating at elastic deformation stage in specimens at foliation angle α = 0° and 90°; (b) lacking stable and unstable crack growth stage for all specimens, especially the specimens under water-saturated condition; (c) dropping rapidly after peak load but maintaining a low residual strength for most specimens. 

(2) The mechanical properties are dependent on foliation angle and water content. The compression strength and peak strain first decrease and then increase with increasing foliation angle in a U-shape or V-shape, and they reach the maximum and minimum values at α = 90° and α = 30°, respectively. The presence of water significantly reduces peak strength and peak strain: The maximum, minimum, and average strength reductions are 71.4%, 10.3%, and 36.0%, respectively; The maximum, minimum, and average strain reductions are 59.5%, 12.6%, and 33.6%, respectively. The elastic modulus varies with the foliation angle in a wavy shape, and it is affected by water content. The effect of water content on elastic modulus is negative for specimens at α = 0°, 30°, and α = 60°, but it is positive for other foliation angles.

(3) With the help of AE, DIC, and SEM techniques, the experiments show that foliation angle and water content have a significant influence on crack evolution. The water acts on biotite and clay minerals that are the main components of phyllite, and it contributes to the initiation, propagation, and coalescence of numerous microcracks. Furthermore, the tensile micro-cracks are the main reason for the failure of the phyllite, especially for saturated groups. From viewpoint of macro-cracks, the initiation shear crack is dominant at lower foliation angles, i.e., 0°, 30°, 45°, and 60°, and the influence of foliation planes on the crack propagation path increases with the foliation angle increasing. At higher foliation angles, such as 90°, tensile cracks initiate and extend along the vertical foliation plane, and finally cause the tensile fracture in foliations from outside to inside.

(4) The ultimate failure modes of foliated phyllite are determined by both foliation angles and water content. Four failure modes are classified as follows: (a) for the specimens with small foliation angles α = 0° and 30° (Water-saturated), both shear sliding and tensile splitting across the foliation planes; (b) for the specimens with low to medium foliation angles α = 30° (Dry) and 45°(Water-saturated), shear sliding dominates the foliation planes; (c) for the specimens with medium to high foliation angles α = 45° (Dry) and 60°, shear sliding dominates the foliation planes; (d) for the specimens with high foliation angles α = 90°, tensile splitting dominates the foliation planes.

## Figures and Tables

**Figure 1 materials-15-08962-f001:**
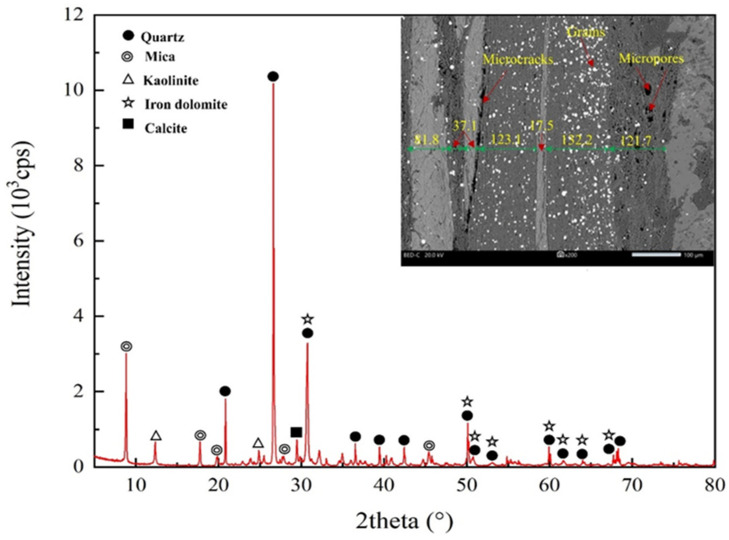
X-ray diffraction (XRD) and Standard Electronic Modules (SEM) analysis.

**Figure 2 materials-15-08962-f002:**
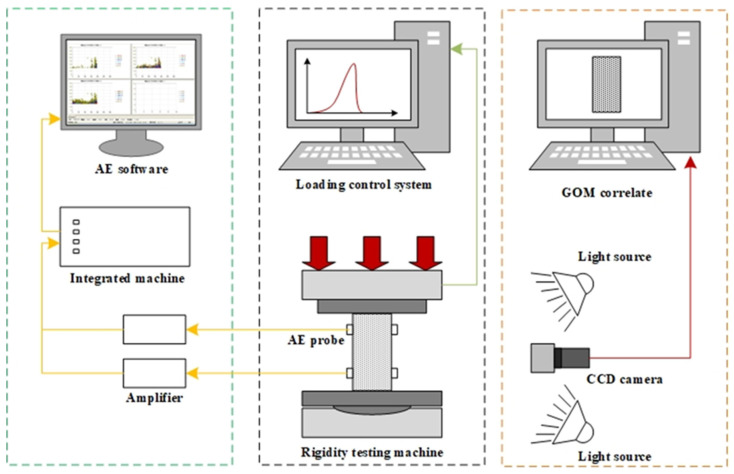
Schematic diagram of the loading system, the DIC system, and the AE system.

**Figure 3 materials-15-08962-f003:**
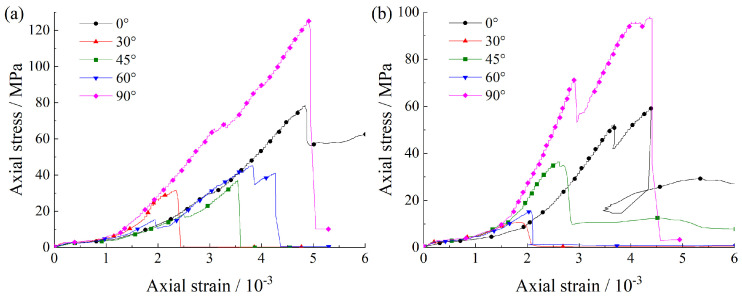
Axial stress-strain curves of layered phyllite with different bedding angles α. (**a**) Dry groups; (**b**) Water-saturated groups.

**Figure 4 materials-15-08962-f004:**
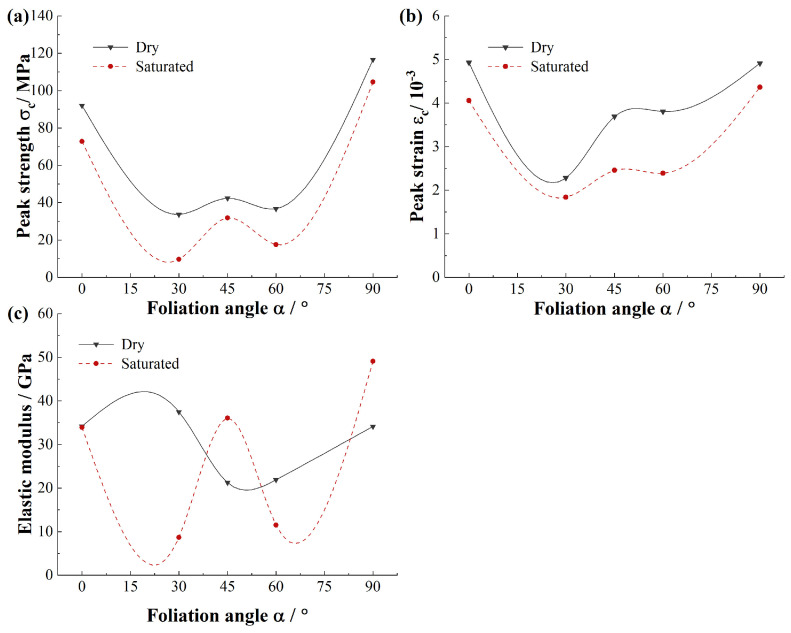
Comparison of mechanical properties of foliated phyllite from dry condition and water-saturated condition: (**a**) Uniaxial compression strength, *σ_c_*; (**b**) Peak axial strain, *ε_c_*; (**c**) Elastic modulus, *E*.

**Figure 5 materials-15-08962-f005:**
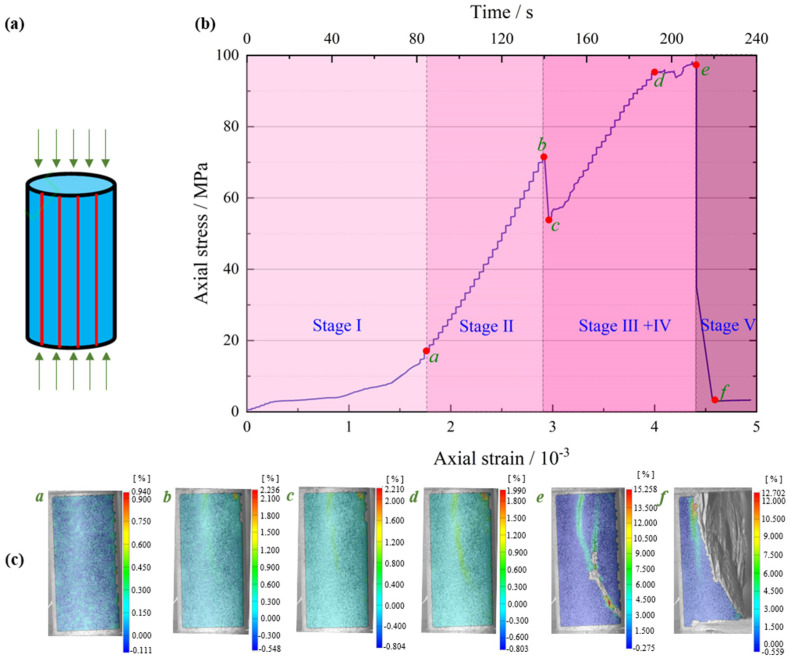
An entire crack evolution process of a typical foliated phyllite subjected to uniaxial compressive loading (S-W-90°): (**a**) Geometrical parameters and loading diagram of the specimen; (**b**) Axial stress–strain curves of a typical specimen; (**c**) The maximum principal strain fields under different loads stage.

**Figure 6 materials-15-08962-f006:**
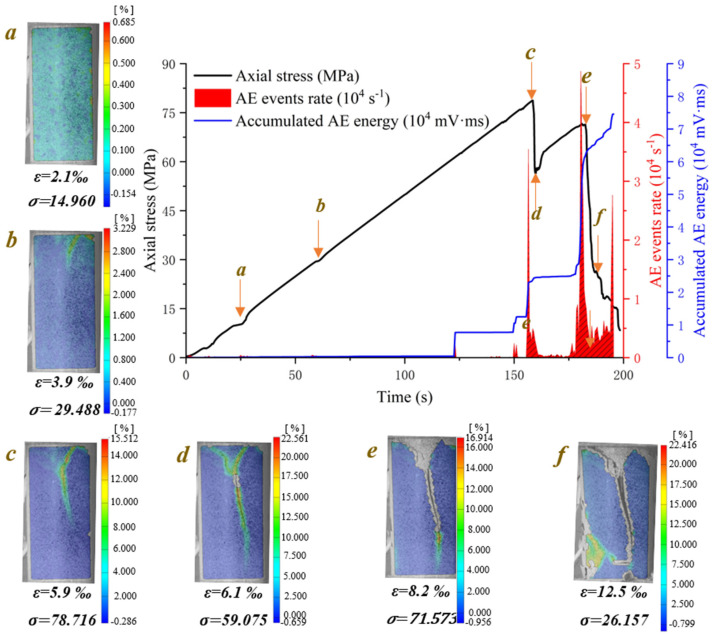
Crack evolution process of dry phyllite with 0° foliation angle (S-D-0°).

**Figure 7 materials-15-08962-f007:**
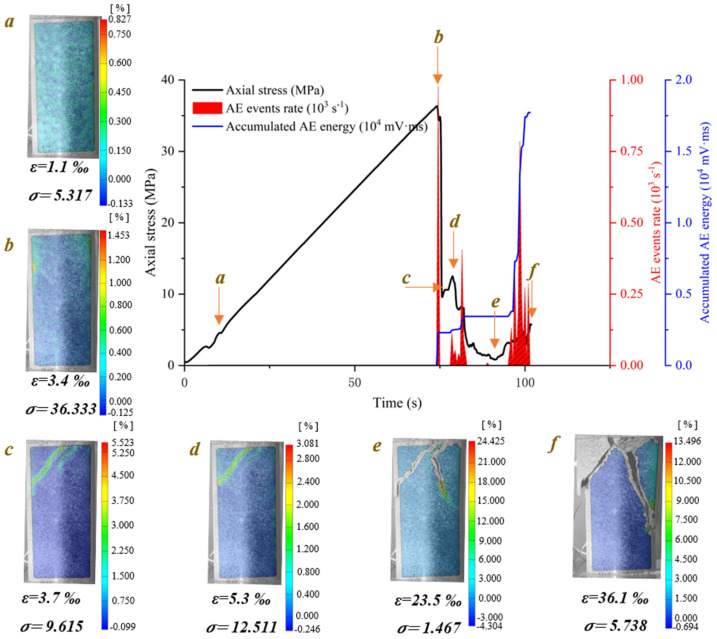
Crack evolution process of water-saturated phyllite with 45° foliation angle (S-W-45°).

**Figure 8 materials-15-08962-f008:**
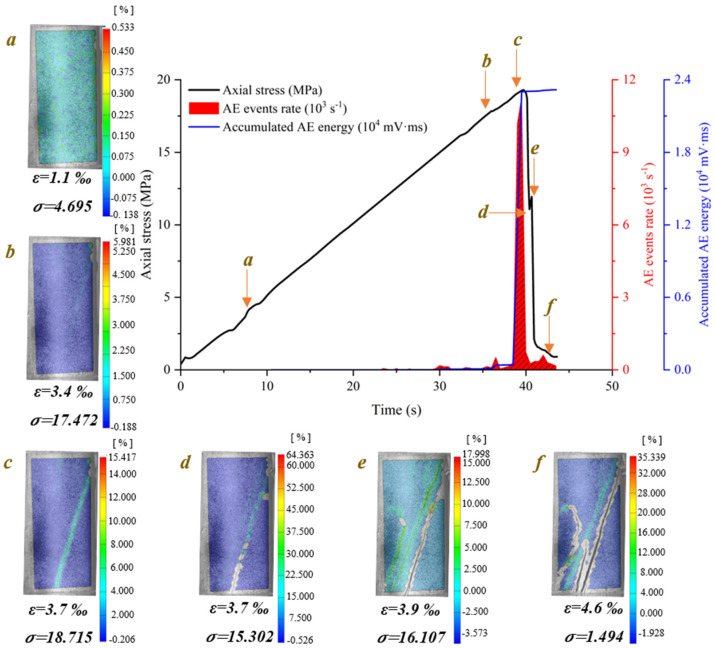
Crack evolution process of water-saturated phyllite with 60° foliation angle (S-W-60°).

**Figure 9 materials-15-08962-f009:**
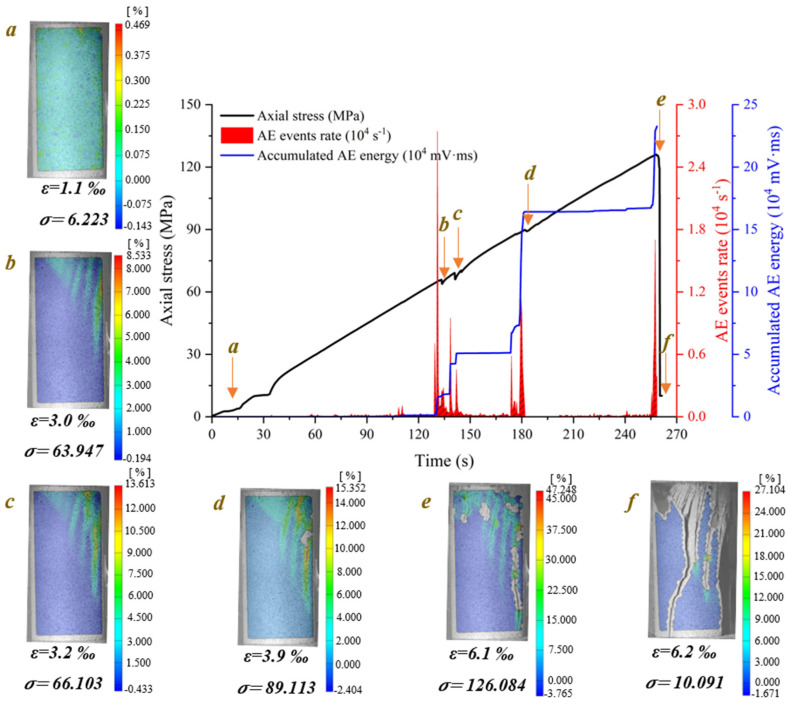
Crack evolution process of dry phyllite with 90° foliation angle (S-D-90°).

**Figure 10 materials-15-08962-f010:**
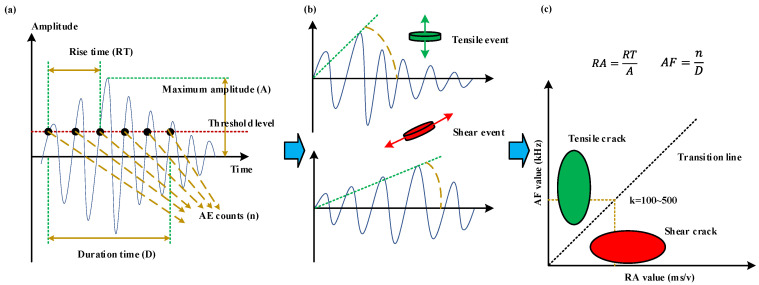
Classification of tensile and shear cracks based on AE parameter analysis: (**a**) Graphical representations of AE characteristic parameters; (**b**) Comparison between typical waveform of tensile event and shear event; (**c**) Crack classification methods in AE parameter analysis.

**Figure 11 materials-15-08962-f011:**
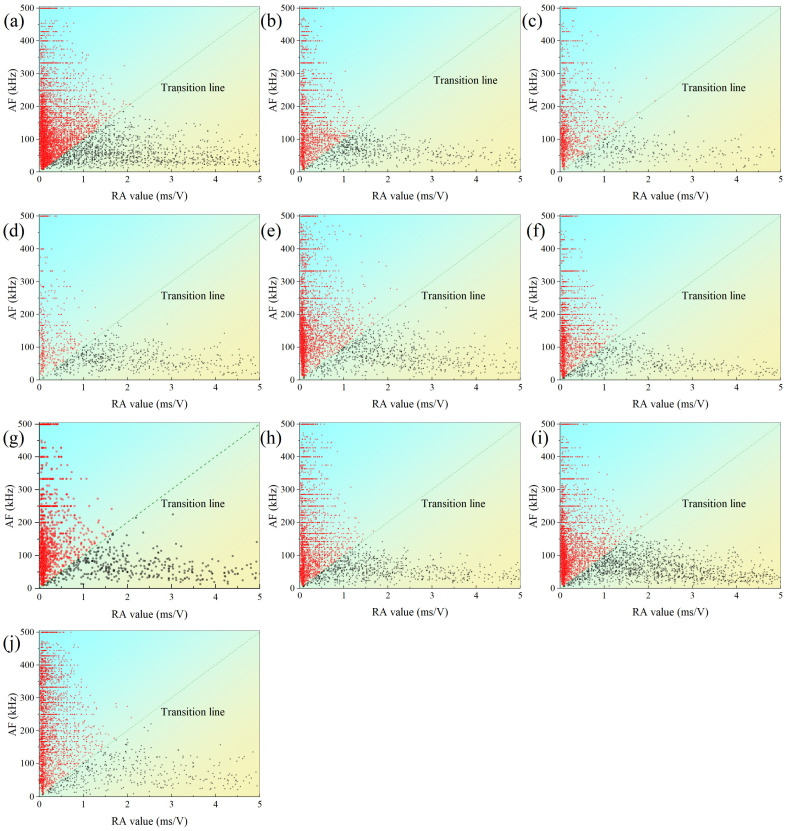
Comparison of AF-RA distribution of foliated phyllite with different foliation angles under dry and water-saturated conditions: (**a**) S-D-0°; (**b**) S-W-0°; (**c**) S-D-30°; (**d**) S-W-30°; (**e**) S-D-45°; (**f**) S-W-45°; (**g**) S-D-60°; (**h**) S-W-60°; (**i**) S-D-90°; (**j**) S-W-90°.

**Figure 12 materials-15-08962-f012:**
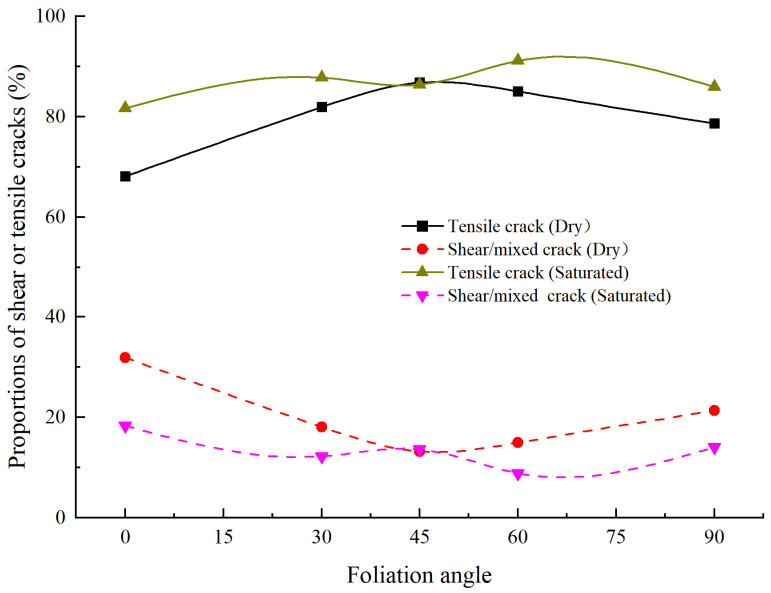
Comparison of the proportions of shear and tensile cracks for foliation phyllite with different foliation angles.

**Figure 13 materials-15-08962-f013:**
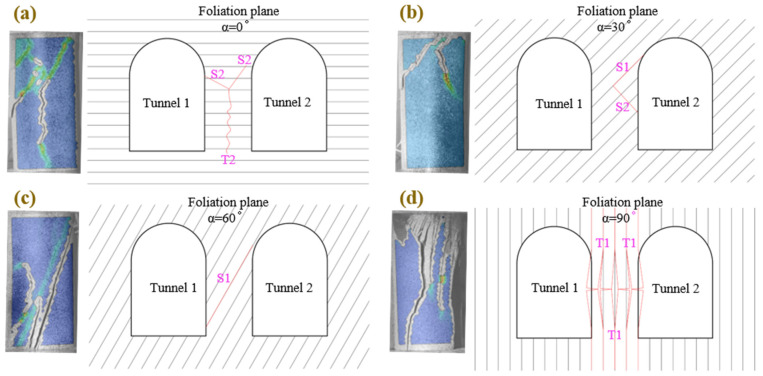
Comparison of failure mechanisms between foliated phyllite with different: (**a**) Shear sliding-tensile splitting across foliation planes (Mode I); (**b**) Shear sliding across foliation planes (Mode II); (**c**) Shear sliding along foliation planes (Mode III); (**d**) Tensile splitting along foliation planes (Mode IV).

**Figure 14 materials-15-08962-f014:**
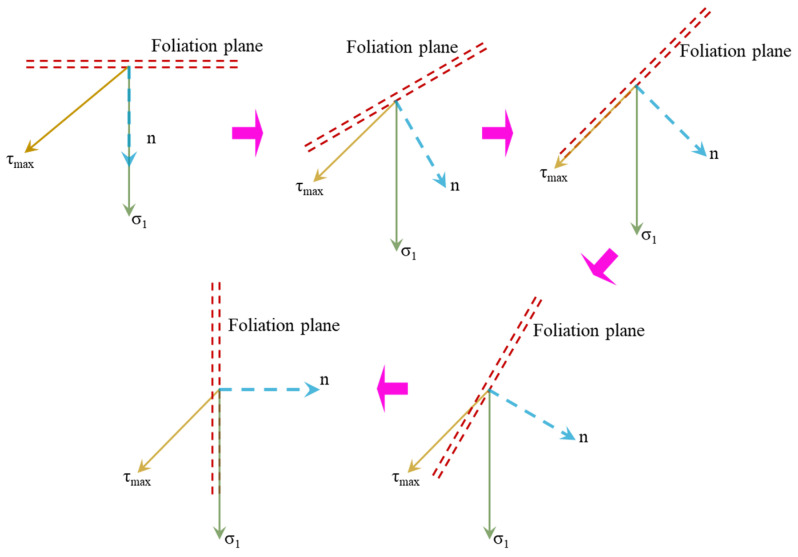
Failure mechanism analysis of foliated phyllite under uniaxial compression.

**Figure 15 materials-15-08962-f015:**
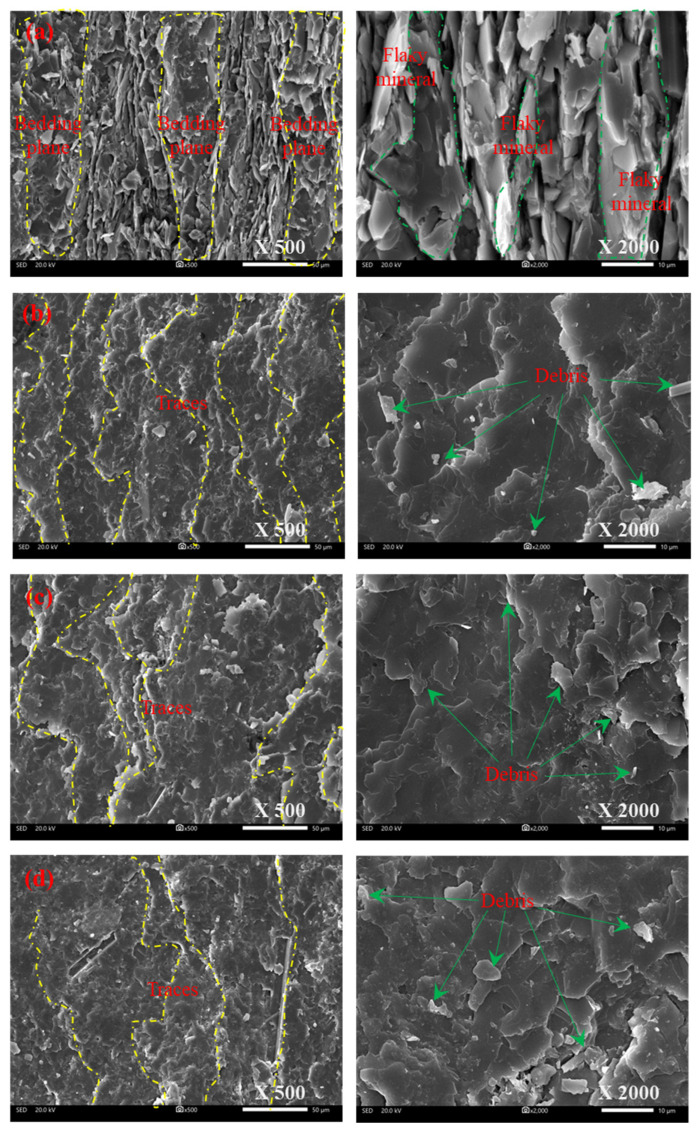
SEM photos of the fracture surfaces for phyllite subjected to shear failure in uniaxial compression: (**a**) S-D-0; (**b**) S-W-45; (**c**) S-D-60; (**d**) S-W-60.

**Figure 16 materials-15-08962-f016:**
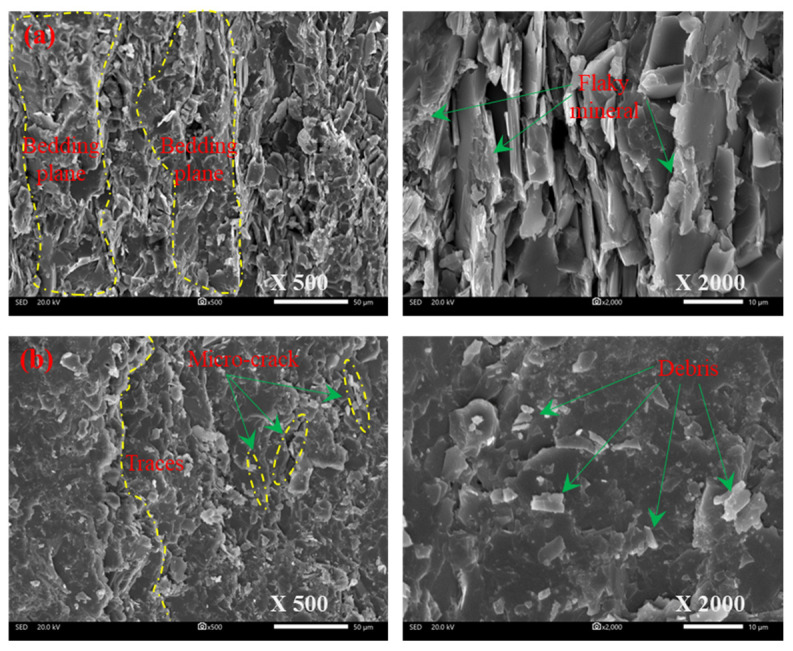
SEM photos of the fracture surfaces for phyllite subjected to tensile failure in uniaxial compression: (**a**) S-D-0; (**b**) S-D-90.

**Figure 17 materials-15-08962-f017:**
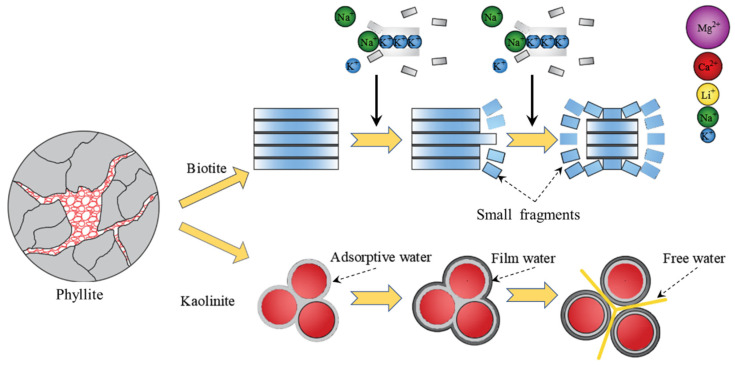
Diagram of the influence of water on phyllite microstructure.

**Figure 18 materials-15-08962-f018:**
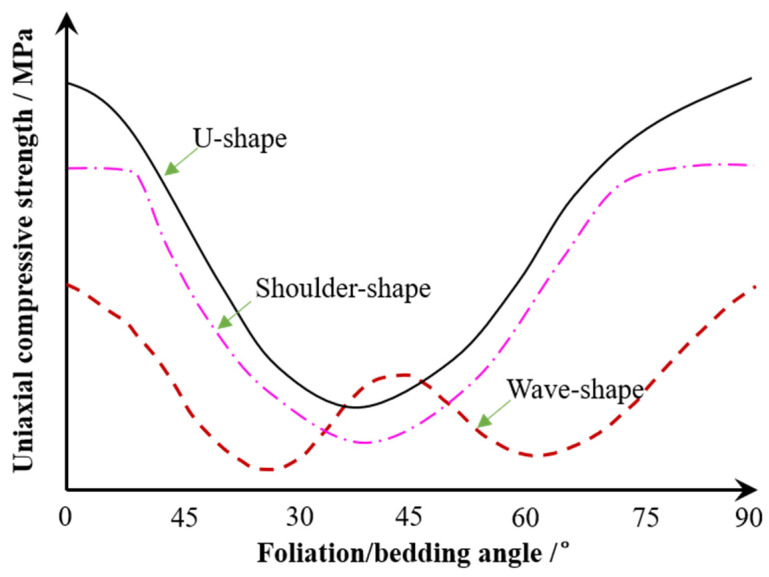
Uniaxial compressive strength of anisotropic rock versus foliation or bedding angle.

**Table 1 materials-15-08962-t001:** Basic physical properties of phyllite with different foliation angles.

Foliation Angle α (°)	Dry Density (g/cm^3^)	Saturation Density (g/cm^3^)	Saturated Water Content (%)	Dry Wave Velocity (m/s)	Saturation Wave Velocity (m/s)
0	2.756	2.775	0.689	3750	4319
30	2.768	2.780	0.434	4332	4573
45	2.762	2.770	0.290	4279	4656
60	2.763	2.782	0.688	4807	5164
90	2.760	2.768	0.290	5942	6095
Average	2.762	2.775	0.471	4622	4961

**Table 2 materials-15-08962-t002:** Ultimate failure modes of foliated phyllite with different foliation angles.

Specimen	Type	0°	30°	45°	60°	90°
Dry groups	Picture	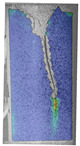	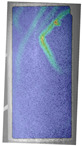	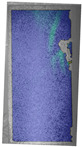	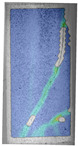	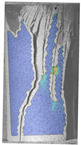
	Forms	Sliding + splitting	Sliding	Sliding	Sliding	Splitting
Saturated groups	Picture	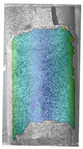	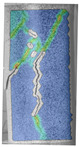	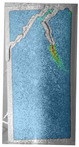	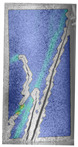	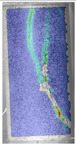
	Forms	Sliding + splitting	Sliding + splitting	Sliding	Sliding	Splitting

**Table 3 materials-15-08962-t003:** Different quantification mineral composition.

Oxides Elements	VD/%	VW/%	PD/%	PW/%
Na_2_O	2.80	3.17	1.70	3.19
MgO	2.70	3.22	2.89	3.94
Al_2_O_3_	13.64	13.65	23.48	21.52
SiO_2_	69.31	71.55	61.71	61.07
K_2_O	3.07	2.46	6.03	4.77
CaO	1.36	1.55	0.26	0.79
TiO_2_	0.61	0.35	1.25	1.54
Fe_2_O_3_	6.50	4.05	2.68	3.18

Notes: VD, vertical fracture surface of foliation plane in specimens under dry condition (S-D-0°); VW, vertical fracture surface of foliation plane in specimens under water-saturated condition (S-W-0°); PD, parallel fracture surface of foliation plane in specimens under dry condition (S-D-60°); PW, parallel fracture surface of foliation plane in specimens under water-saturated condition (S-W-60°).

**Table 4 materials-15-08962-t004:** Classification of ultimate failure mode in the present experimental results.

Type	Label	Description of Failure Modes Type	Failure Modes	Specimens
I	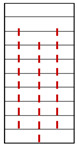	Tensile fracture across discontinuities	T2	None
II	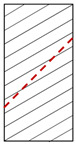	Sliding failure across discontinuities	S2	None
III	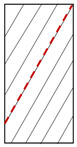	Sliding failure along discontinuities	S1	S-D-45 S-D-60 S-W-60
IV	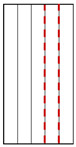	Tensile-split along discontinuities	T1	S-D-90 S-W-90
V	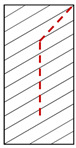	Sliding failure across discontinuities and Tensile fracture across discontinuities	M1 (S2 + T2)	S-D-0 S-W-0 S-W-30
VI	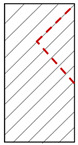	Sliding failure along discontinuities and Sliding failure across discontinuities	M2 (S1 + S2)	S-D-30 S-W-45

## Data Availability

The data used to support the findings of this study are available from the corresponding author upon request.

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
