# Peer review of "Mechanical Properties and Failure Behavior of Dry and Water-Saturated Foliated Phyllite under Uniaxial Compression"

_materials, 2022, doi:10.3390/ma15248962_

Round 1

Reviewer 1 Report

Dear Authors,

In this research, the author provided an experimental study to investigate mechanical properties and failure behavior of foliated phyllite in dry and saturated states under uniaxial compression. The subject of this paper is interesting and in general, the manuscript is well written and organized. The essential characteristics of the research program are reported and the figures are carefully prepared. But the manuscript needs some further improvements before to be accepted for publication. There are some grammatical errors through the manuscript. The reviewer has listed some specific comments that might be helpful for the authors to enhance the quality of the manuscript as bellow:

1.    The words “layer” “layered” or “bedding” must be replaced by suitable words such as “foliation”, “anisotropic” or “foliated” throughout the text. In fact, layer and bedding are used for sedimentary rocks and foliation for metamorphic rocks in academic texts. Phyllite is a metamorphic rock.

2.    The abstract is well written.

3.    The introduction is detailed. But it needs a significant amount of reorganization by referring to previous research recent works and new references. The authors can see the bellow references for more information:

10.1520/GTJ20130078

https://doi.org/10.1007/s00603-015-0814-y

4.    For readers to quickly catch your contribution, it would be better to highlight major difficulties and challenges, and your original achievements to overcome them, in a clearer way in abstract and introduction.

5.    Sampling, specimen preparation, and test procedures or methods must be referenced to ASTM or ISRM suggested methods in “Materials and Methods”. For example, ISRM (2007) ... Also, all references have to add to references list.

6.    In Table 1, are the values of saturation relevant to the natural state? If yes, please note in the bottom of the table as a foot note.

7.     In Fig. 20, the concepts of a-e must be written in the figure caption.

8.    In general, Discussion is well written. Nevertheless, the discussion should provide a summary of the main finding(s) of the manuscript in the context of the broader scientific literature, as well as addressing any limitations of the study or results that conflict with other published work.

9.    The “Conclusions” reports methodology and some results of the research. It is provided in an ideal state.   

10. Please check and improve the English throughout the paper. Some sentences are hardly understandable. The English language of the paper needs to be improved.

Best Wishes

The end

Author Response

We have provided a point-by-point response to your comments with a upload PDF file.

Reviewer 2 Report

The work itself is well done. The relevance of the work is low as similar tests for other kinds of rocks exist all over the literature and transfer of the results into real-world examples is barely possible due to the small lab scale, only uni-axial compression, natural heterogeneity, etc.

Major comments:

* The paper needs restructuring: Several parts in the discussion belong into the results chapter. In the Discussion no new results should be presented (here: SEM photos)

* Fig. 7 to 15 are quite repetitive. Consider making a supplement for these figures and only discuss one of these figures within the paper in detail and discuss the differences with respect to this sample. Not necessary but might benefit the flow of the paper.

* Fig. 19 needs re-working. It is hard to understand and the description is lacking information (What is A & B? What is symboled by the lines?).

* The experimental description is lacking information. How were the values in Table 1 determined? How was the sample kept saturated (leather jacked?)?

* The manuscript still contains sentences from the template. 

* The literature review about the current state is incomplete. I recommend the authors to also consider Non-Chinese authors (4 out of 37 literature sources include authors from institutions outside of China).

Minor comments and more details are given in the attached pdf.

Author Response

(The authors gave the same response as above.)

Round 2

Reviewer 2 Report

The authors addressed all my comments and especially modified their discussion which clearly benefits the manuscript! I recommend acceptance.